# Genetic diversity of *Cryptosporidium* spp. in non-human primates in rural and urban areas of Ethiopia

**Ambachew W. Hailu**[1]*, **Abraham Degarege**[2], **Beyene Petros**[1], **Damien Costa**[3,4], **Yonas Yimam Ayene**[5], **Ven-ceslas Villier**[3], **Abdelmounaim Mouhajir**[3], **Loic Favennec**[3,4], **Romy Razakandrainibe**[3,4☯], **Haileeysus Adamu**[6☯]

**1** Department of Microbial Cellular and Molecular Biology, Biomedical Sciences Stream Addis Ababa University, Addis Ababa, Ethiopia, **2** Department of Epidemiology, College of Public Health, University of Nebraska Medical Center, Omaha, Nebraska, United States of America, **3** EA ESCAPE 7510, University of Medicine Pharmacy Rouen, Rouen, France, **4** CNR LE Cryptosporidiosis, Santé Publique France, Rouen, France, **5** Department of Medical Parasitology and Mycology, School of Public Health, Tehran University of Medical Sciences, Tehran, Iran, **6** Institute of Biotechnology, Addis 12 Ababa University, Addis Ababa, Ethiopia

☯ These authors contributed equally to this work.
* ambalake@gmail.com

**Data Availability Statement:** All relevant data are within the manuscript.

**Funding:** The authors received no specific funding for this work.

## Abstract

Non-Human Primates (NHPs) harbor *Cryptosporidium* genotypes that can infect humans and vice versa. NHPs *Chlorocebus aethiops* and *Colobus guereza* and humans have over-lapping territories in some regions of Ethiopia, which may increase the risk of zoonotic transmission of *Cryptosporidium*. This cross-sectional study examined the molecular prevalence and subtypes of *Cryptosporidium spp.* from 185 fecal samples of *Chlorocebus aethiops* and *Colobus guereza* in rural and urban areas in Ethiopia. Samples were tested for *Cryptosporidium* infection using nested polymerase chain reaction (PCR), and subtypes were determined by sequencing a fragment of the 60-kDa glycoprotein gene (gp60). Of the 185 samples, fifty-one (27.56%) tested positive for *Cryptosporidium* infection. The species detected were *C. parvum* (n = 34), *C. hominis* (n = 12), and *C. cuniculus* (n = 3). Mixed infection with *C. parvum* and *C. hominis* were detected in 2 samples. Four *C. hominis* family subtypes (Ia, Ib, Id, and Ie) and one *C. parvum* family subtype (IIa) were identified. *C. hominis* IaA20 (n = 7) and *C. parvum* IIaA17G1R1 (n = 6) were the most prevalent subtypes detected. These results confirm that *Chlorocebus aethiops* and *Colobus guereza* can be infected with diverse *C. parvum* and *C. hominis* subtypes that can also potentially infect humans. Additional studies could help to understand the role of NHPs in the zoonotic transmission of *Cryptosporidium* in Ethiopia.

## Introduction

Cryptosporidiosis is a zoonotic parasitic disease that cause diarrhea in humans, farm animals and nonhuman primates (NHPs); water and foodborne outbreaks [1,2]; and malnutrition and

**Competing interests:** The authors have declared that no competing interests exist.

cognitive deficits in children [3]. Cryptosporidiosis causes considerable morbidity and mortality in untreated acquired immunodeficiency syndrome (AIDS) patients and production loss in developing countries [4,5]. Globally, over 550 water-borne and foodborne outbreaks have been linked to cryptosporidiosis [6,7].

NHPs coexist with humans and livestock in an anthropogenically altered habitat [8,9]. This spatial proximity and close interaction between humans and NHPs may increase the risk of potential cross-species *Cryptosporidium* transmission [10–12]. Indeed, several species of *Cryptosporidium* that infects humans have also been identified from NHPs in Asia, America, and Africa [13–15]. In western Uganda, potential cross-species *Cryptosporidium* transmission between humans and NHPs was documented [16]. Similarly, a study conducted in Tanzania also highlighted the potential of *Cryptosporidium* for cross-species transmission between humans and NHPs [17]. Studies conducted in Rwanda, China, and Thailand also reported eight *Cryptosporidium* species including, *C. hominins*, *C. parvum*, *C. felis*, *C. muris*, *C. ubiquitum*, *C. meleagridis*, *C. bovis*, and *C. andersoni* in NHPS which can also infect human [14,18,19].

*Chlorocebus aethiops* (vervet monkey) and *Colobus guereza* (mantled guereza) are the most common NHPs in Ethiopia and they coexist in shared habitats with humans and other animals [20,21]. While vervet monkeys are regarded as a nuisance in some Ethiopian regions, mainly due to crop-raiding activities, mantled guereza is known for their cohabitation behavior with human settlement adjoining the forest areas in the southern part of Ethiopia [22,23]. Hence, identifying genotype and subtype of *Cryptosporidium* in NHPs is of paramount importance to understand their role as a reservoir of infections to humans and other animals. However, studies on the prevalence and distribution of *Cryptosporidium* species/genotype/subtypes in NHPs and their zoonotic potential are scanty. As there are no vaccines to prevent infection with *Cryptosporidium* and there are limited drug to treat infection with this parasite [1], accurate identification of species and subtypes of *Cryptosporidium* in vast arrays of host species, including NHPs is crucial for the characterization of transmission patterns and potential control options. We, therefore, carried out this study to shed new light on the genetic diversity of *Cryptosporidium* infection in NHPs in Ethiopia.

## Methods

### Study area description

The study was conducted in urban (i.e., a recreational site on the eastern shore of Lake Hawassa city) and rural areas (small village called Wurgissa) from June through September 2018. Subjects were free-ranging wild vervet and colobus monkeys found in two localities. Hawassa urban area offer colobus and vervet monkeys easy access for interacting with visitors, and people in the open recreation area often feed the monkeys with leftover food. However, only vervet monkeys were found in the rural area and participated in this study from the vervet monkeys in the rural site move throughout the village, spending substantial amounts of time feeding, defecating, and sleeping near the human villagers, and picking food off the ground contaminated with cattle faeces.

### Stool sample collection and identification of NHPs

Verbal approval was obtained from the residents of Wurgissa and Hawassa, who live near the sampling site, and from local authorities (Hawassa city agriculture office and Wurgissa Kebele administration) before collecting samples from the NHPs. A single fresh fecal sample was collected from each monkey's feces dripping off the floor following standard noninvasive sampling procedure [24]. Briefly, when a NHPs troop were encountered, first the total number of

monkeys in a troop, the species and their geospatial location, age (adult, infant) and sex (male, female) information were recorded and followed until they defecate. Then the monkeys in the troop were followed until they defecate or lost from the connection. When monkeys defecated, the appearance of the faeces identified (diarrheic, formed), and an approximately 1 gram placed into 100ml plastic containers and mixed thoroughly with a 2.5% potassium dichromate solution. Samples within the plastic containers were stored at +4˚C and transported to the bio-medical laboratory of the Addis Ababa University. At the end, samples transferred to the laboratories of expert Cryptosporidiosis at the Charles Nicole University Hospital in France.

## DNA extraction, molecular detection, and subtyping

Before the DNA isolation was performed, the stool samples had their preservative (2.5% potassium dichromate solution) removed in centrifugation process at 3000 X $g$ for 10min at 4˚C with PBS (pH = 7.2) repeated thrice. Total genomic DNA from each fecal sample was extracted using QIAamp Power Fecal DNA Kit (Qiagen). DNA extraction was performed with 250 µL of the fecal sample in accordance with the manufacturer's instructions. Eluted DNA was stored at −20˚C prior to PCR analysis [25]. Sequence-based characterization of Cryptosporidium has been performed to determine species and genotype. *Cryptosporidium* species were detected by nested PCR amplification of the SSU rRNA gene as described by Koehler et al., 2017 [26], PCR amplifications were performed with positive (*C. parvum and C. hominis*) and negative controls (no DNA water). PCR products were visualized on a UV transilluminator following electrophoresis on 2% agarose gels stained with ethidium bromide. All samples positive for *Cryptosporidium* species at the 18S locus were further subtyped at the 60 kDa glycoprotein (gp60) locus using a nested-PCR producing a ~ 364 bp secondary product, as previously described elsewhere [26]. The amplified DNA from secondary PCR products was separated by gel electrophoresis and sequenced using an ABI3500 sequencer analyzer (Applied Biosystems, Foster City, CA). Sanger sequencing chromatogram files were imported into Bioedit (http://www.mbio.ncsu.edu/BioEdit/bioedit.html), edited, analyzed, and aligned with reference sequences from GenBank.

## Statistical analysis

Statistical analysis was performed using IBM SPSS Statistics software (version 26). Chi-square test was used to test the association of *Cryptosporidium* infection with the age (adult *vs* infant), sex (male *vs* female), species (vervet *vs* colobus), stool appearance (non-diarrheic *vs* diarrheic), and location (rural *vs* urban) of the NHPs. Because the number of *Cryptosporidium* infected cases in colobus monkey, and non-Diarrheic category were less than 5 infections, fisher exact test was used to test the association of Cryptosporidium infection with species and stool appearance used for cell values less than 5. P values < 0.05 were considered statistically significant.

## Ethic statement

This study was part of another study which was approved by the ethical review board of Addis Ababa university. permission to collect fecal samples was obtained from the residents of Wurgissa and Hawassa, who live near the sampling site, and from local authorities (Hawassa city agriculture office and Wurgissa Kebele administration) before collecting samples from the NHPs.

## Results

### Prevalence of *Cryptosporidium* in NHPs

Faeces from 185 NHPs (177 *Chlorocebus aethiops* and 8 *Colobus guereza*) located in rural Wurgissa (n = 145) and urban Hawassa (n = 40) were examined. Out of the 185 NHPs, fifty-one

**Table 1. Distribution of *Cryptosporidium* species infection in NHPs from Wurgissa and Hawassa areas in Ethiopia and their geospatial localization.**

| Areas | Troop s ID | Geospatial location | C.cuniculus | | C.hominis | | C.parvum | | C.hominis / C.parvum | | Negative | | Total |
|---|---|---|---|---|---|---|---|---|---|---|---|---|---|
| | | | *Vervet* | *Colobus* | *Vervet* | *Colobus* | *Vervet* | *Colobus* | *Vervet* | *Colobus* | *Vervet* | *Colobus* | |
| **Urban area** | HA | Amora Gedel park (7°02'37.4"N 38°27'24.1"E) | | | | 1 | 10 | 3 | | | 8 | 4 | 26 |
| | HB | Wabishebel Hotel (7°02'55.6"N 38°27'34.7"E) | | | 1 | | 2 | | | | 11 | | 14 |
| **Rural area** | WA | Gollo (11°32'22.1"N 39°36'28.7"E) | | | 2 | | 2 | | | | 15 | | 19 |
| | WB | Gorarba (11°32'10.1"N 39°40'45.4"E) | 1 | | 3 | | 2 | | | | 11 | | 17 |
| | WC | Worekalu (11°34'15.1"N 39°40'03.2"E) | | | | | 4 | | 1 | | 15 | | 20 |
| | WD | Gatira Georgis church (11°33'10.2"N 39°36'51.6"E) | 2 | | 2 | | 2 | | 1 | | 26 | | 33 |
| | WE | Goda (11°31'55.4"N 39°37'18.9"E) | | | 3 | | 2 | | | | 14 | | 19 |
| | WF | Burka (11°33'21.6"N 39°39'39.6"E) | | | | | 6 | | | | 15 | | 21 |
| | WG | Gebriel (11°32'43.8"N 39°37'13.9"E) | | | | | 1 | | | | 15 | | 16 |
| | Total | | 3 | | 11 | 1 | 31 | 3 | 2 | | 134 | | 185 |

(51) were infected with *Cryptosporidium* species based on a nested PCR test (**Table 1**). The prevalence of *Cryptosporidium* infection was significantly higher in female (35.4%) than in male (14.70%) monkeys($p = 0.002$). A significantly greater proportion of samples collected from monkeys in urban (i.e., Hawassa, 42.5%) area were positive for *Cryptosporidium* compared to samples obtained from monkeys in the rural area (i.e., Wurgissa, 23.44%) ($P = 0.017$). In terms of NHPs species, *Cryptosporidium* infection was detected in 47 out of 177 *Chlorocebus aethiops* and 4 out of 8 *Colobus guereza*. The monkeys were asymptomatic carriers of *Cryptosporidium* ($p = 0.001$). However, the prevalence of *Cryptosporidium* infection did not vary with the age of the monkey (Table **2**).

### *Cryptosporidium* species and genotypes distribution

Sequence analysis of 51 positive samples from Gp60 and 18s RNA gene revealed three species: *C. parvum* (n = 34), *C. hominis* (n = 12) and *C. cuniculus* (n = 3). Two samples with mixed infections of *C. hominis* and *C. parvum* were detected. Out of the 51 positive samples, 16 of the 34 *C. parvum* and 11 of 12 *C. hominis* positive isolates were successfully subtyped. Diverse subtype families, including IIa, Iba, IId and Ie, were seen in the rural study area, but only the IIa subtype family was identified in the urban area.

**Table 2. Prevalence of *Cryptosporidium* infection by age, sex, and stool character among NHPs.**

| Attributes | Categories | *Cryptospridium* infection rate SSU-rRNA PCR % (n/N) | P-value* | *Cryptospridium* spp (n) | | | | Subtype (n) |
| --- | --- | --- | --- | --- | --- | --- | --- | --- |
| | | | | *C. parvum* | *C. hominis* | *C. cuniculus* | Mixed | |
| Prevalence (n/N) | | 27.56 (51/185) | | | | | | |
| Location | Wurgissa (rural) | 23.44 (34/145) | 0.017 | 19 | 10 | 3 | 2 | IIaA17G1R1(6), IIaA19G2R1(3), IIaA15G2R1(2), IIaA16G2R1(1) IIaA16G1R1(1) IIaA17G2R1(1), IIaA20G1R1(1), and IaA20 (5), IdA21(1) lbA10G2(1), IeA11G3T3(1), IaA26 (1) |
| | Hawassa (urban) | 42.50 (17/40) | | 15 | 2 | | | IaA20(2), IIaA15G2R1(1), |
| Sex | Male | 14.70 (10/68) | 0.002 | 6 | 2 | 1 | 1 | IIaA19G2R1(1), IIaA16G2R1(1), lbA10G2 (1) |
| | Female | 35.04 (41/117) | | 28 | 10 | 2 | 1 | IIaA17G1R1(6), IIaA19G2R1(2), IIaA15G2R1(3), IIaA16G1R1(1), IIaA17G2R1(1), IIaA20G1R1(1), IaA20(7), IdA21(1), IeA11G3T3(1) IaA26(1) |
| Age group | Infants | 23.52 (20/85) | 0.26 | 12 | 4 | 2 | 2 | IIaA17G1R1(3), IIaA16G1R1(1) IIaA15G2R1(1), IIaA20G1R1(1), IaA20(3), lbA10G2(1), IaA26 (1), |
| | Adult | 31 (31/100) | | 22 | 8 | 1 | | IIaA17G1R1 (3), IIaA19G2R1(3), IIaA17G2R1(1), IIaA15G2R1(2), IaA20 (4), IdA21(1), IeA11G3T3(1) |
| Species | colobus | 50(4/8) | 0.218 | 3 | 1 | | | IaA20(1) |
| | vervet | 26.6(47/177) | | 31 | 11 | 3 | 2 | IIaA17G1R1 (6), IIaA19G2R1 (3), IIaA15G2R1(3) IIaA16G1R1 (1), IIaA17G2R1(1), IIaA20G1R1(1) (IaA20 (6), IaA26 (1), lbA10G2 (1), IdA21 (1) IeA11G3T3 (1) |
| Appearance of the stool | Diarrheic | 0 (0/32) | 0.001 | | | | | |
| | Non-Diarrheic | 33.33 (51/153) | | 34 | 12 | 3 | 2 | IIaA17G1R1 (6), IIaA19G2R1 (3), IIaA15G2R1(3) IIaA16G1R1 (1), IIaA17G2R1(1), IIaA20G1R1(1) IaA20 (7), IaA26 (1), lbA10G2 (1), IdA21 (1) IeA11G3T3 (1) |

Subtyping analysis of the *C. parvum* and *C. hominis* isolates from *Chlorocebus aethiops* identified five family subtypes for *C. hominis* {IaA20 (n = 6), IaA26 (n = 1); lbA10G2 (n = 1), IdA21 (n = 1) IeA11G3T3 (n = 1)}, and six family subtypes for *C. parvum* {IIaA17G1R1 (n = 6), IIaA19G2R1 (n = 3), IIaA15G2R1(n = 3) IIaA16G1R1 (n = 1), IIaA17G2R1(n = 1), IIaA20G1R1(n = 1)}. Subtyping analysis of the four *Cryptosporidium* positive samples obtained from *Colobus guereza* revealed the subtype 'IaA20'. A total of 12 subtypes {IIaA17G1R1(6), IIaA19G2R1(3), IIaA15G2R1(2), IIaA16G2R1(1) IIaA16G1R1(1) IIaA17G2R1(1), IIaA20G1R1(1), IaA20 (5), IdA21(1) lbA10G2(1), IeA11G3T3(1), IaA26(1))} were seen from NHPs located in rural Wurgissa but only two subtypes {IaA20 (2), IIaA15G2R1 (1)}were identified from monkeys in the Hawassa town (Table 2).

## Discussion

There is limited information about the molecular prevalence and diversity of *Cryptosporidium spp*. in NHPs in Ethiopia. This study examined the molecular prevalence of *Cryptosporidium*

*spp.* subtypes in *Chlorocebus aethiops* (vervet monkey) and *Colobus guereza* (mantled guereza) in Ethiopia. Out of 185 NHPs samples examined, *C parvum*, *C. hominis* and *C. cuniculus* were found in 34, 12 and 3 samples, respectively. Amongst multiple subtype families identified, *C. hominis* IaA20 was the most frequently seen.

The prevalence of *Cryptosporidium* infection among vervet monkeys (47/177, 26.6%) detected in the current study area was higher than the prevalence reported among vervet monkeys in other regions of Ethiopia (3.5% to 9.5%) and Tanzania (16%) [17,20]. A study in Indonesia (2.7%) and Central African Republic (0.5%) also reported lower prevalence of *Cryptosporidium* infection in *Gorilla beringei* (Mountain gorillas) [11,27]. However, a study in Uganda showed a higher prevalence of *Cryptosporidium spp.* among free-ranging mountain gorillas (73%, 8/13) [28]. The variation in the prevalence of infection in the different regions could be due to the difference in the NHPs hosts examined; encroachment habits of NHPs to humans and domestic animals; presence of habitats for feeding and ingestion of water. The sensitivity and specificity of the diagnostic test used could also contribute to the differences in the magnitude of the prevalence of infection reported by the studies. Some of the studies used microscope to check infection [19,25] which is less sensitive compared to the molecular methods applied in the current study. The size of the sample examined could also be a source of variation in the prevalence of infection reported in the studies. Most of these studies involved a sample size of lower than 100 [15,19,25,29] and even some lower than 20 samples [20] compared to the 185 samples examined in the current study.

In agreement with the previous studies, *C. hominis* and *C. parvum* have been detected in NHPs [13,15,30–32]. These reports suggest that *C. hominis*, which was initially described as infectious to humans, may potentially expand its hosts from humans to NHPs. Indeed, *C. hominis* has been reported in several domestic livestock, wildlife hosts, rabbits, other mammals (marsupials) [33–37]. Among the NHPs examined, *C. cuniculus* was also detected in three samples. To the best of our knowledge, *C. cuniculus* has not been previously reported in NHPs. However, the zoonotic potential of *C. cuniculus* was apparent when it was responsible for a drinking-water-associated outbreak of cryptosporidiosis in the United Kingdom [38]. The likely hood of an increase in close contact between humans and NHPs due to the incursion of NHPS into agricultural fields, homes, and recreational areas may contribute to the increased prevalence of *C. hominis* infections in NHPs in Ethiopia.

This study documented multiple subtype families of *C. hominis* including Ia, Ib, Id, and Ie. Studies conducted in China have also shown diverse *C. hominis* subtype families including Ia, Id, Ie and If in NHPs [13,30]. Also, a survey in Kenya showed subtype families of *C. hominis* including Ib, If and Ii [39]. In Ethiopia, *C. hominis* subtype families Id, Ie, and Ib have been identified in humans [40]. Due to some level of similarity in their genetics make up, humans and NHPs could probably be susceptible to the same pathogens, including *C. hominis* [41]. Moreover, the incursion behaviors of non-human primates to the human habitat may enhance the probabilities of spread through faeces contaminated food and water.

*C. parvum* IIaA17G1R1 (n = 6) was the most common subtype found in this study. This subtype has been reported in human samples in Wales and England [42,43], Romania [44], Iranian [45], The Netherlands [46] suggesting that the subtype has a potential to expand its hosts from humans to NHPs. This could be linked to the subtype's infectivity, pathogenesis, transmissibility, and host adaption potential, which are shaped by genetic re-combination and selective pressure of the *C. parvum* population structure [1]. The predominant isolate of *C. parvum* (IIaA17G1R1) was frequently detected in the rural area. Domestication of livestock, animal manure, or excreta as a fertilizer, poor latrine uses, and conducive temperature and rainfall patterns infection may increase the likelihood of fecal-oral transmission *C. parvum* in rural areas. The transmission dynamic remains obscure, reinforcing the need for multi-locus

genotype analysis of the parasite (isolated from human, livestock, and water samples) to properly elucidate the host population structure and the dynamic of *Cryptosporidium* transmission.

A higher prevalence of *Cryptosporidium* infection was seen among NHPs living in the urban area compared to the NHPs living in the rural village. This difference might have resulted from ecologic factors (e.g., fecal contaminated water and close habitations), which may facilitate transmission of the parasite between NHPs and humans. Similarly, the higher prevalence *Cryptosporidium* infection in adult monkeys might have been related to greater longevity, increasing the risk of acquiring *Cryptosporidium* infection from contaminated drinking water and contaminated food.

A significantly higher prevalence of *Cryptosporidium* infection was seen in females than males NHPs. Studies in Sri-Lanka [47] and China [30] also reported an increased prevalence of *Cryptosporidium spp.* in female NHPs than males. This observed difference in the prevalence could be due to a difference in the risk of infections between males and females. Male and female NHPs may not contact with humans and other domestic animals and ingest fecal contaminated water equally. Male and female NHPs may also have different preferences in their dwelling area, affecting susceptibility to *Cryptosporidium* infection.

*Cryptosporidium* species can infect a broad range of hosts, including humans, domestic and wild animals worldwide, causing asymptomatic or mild to severe gastrointestinal disease in their host species. In this study, all the *Cryptosporidium* infected NHPs were asymptomatic. Although *Cryptosporidium* infections can lead to watery diarrhea in infected hosts, *no Cryptosporidium* parasite was detected among 32 diarrheic stool samples. The absence of *Cryptosporidium* infections in diarrheic stool samples could be explained by the varying degree of pathogenicity and virulence isolates of the same species of *Cryptosporidium*, replication of the parasite, and the resulting immune response [48].

This study is not without limitations. Although the nested PCR method has been described as very sensitive and specific [49], out of the 51 *Cryptosporidium*-positive samples identified at SSUrRNA level, Gp60 PCR amplification and sequencing failed during this study for 25 samples. This lower efficiency of PCR for Gp60 (as compared to 18S rRNA gene) is due to the lower gene copy number and longer length of the amplified fragment [25,50]. In addition, sequence analysis was unsuccessful for *C. cuniculus* due to lower gene copy number and longer length of the amplified fragment.

In conclusion, findings from this study suggest that *Chlorocebus aethiops* and *Colobus guereza* can be infected with diverse *C. parvum* and *C. hominis* subtypes, which are likely associated with human infection. Additional studies involving the characterization of parasites from livestock, drinking, and recreational water sources could help understand the direct or indirect transmissions between NHPs, livestock, and humans and elucidate the role of NHPs in the zoonotic transmission of *Cryptosporidium*.

## Acknowledgments

The authors are grateful, Rouen University Hospital (France), and Addis Ababa University (Ethiopia) support.

## Author Contributions

**Conceptualization:** Ambachew W. Hailu, Beyene Petros, Haileeysus Adamu.

**Data curation:** Ambachew W. Hailu, Abraham Degarege, Yonas Yimam Ayene, Abdelmounaim Mouhajir, Romy Razakandrainibe, Haileeysus Adamu.

**Formal analysis:** Ambachew W. Hailu, Abraham Degarege, Yonas Yimam Ayene, Abdelmou-naim Mouhajir, Romy Razakandrainibe, Haileeysus Adamu.

**Funding acquisition:** Ambachew W. Hailu, Beyene Petros, Loic Favennec.

**Investigation:** Ambachew W. Hailu, Damien Costa, Ven-ceslas Villier, Loic Favennec, Romy Razakandrainibe, Haileeysus Adamu.

**Methodology:** Ambachew W. Hailu, Abraham Degarege, Beyene Petros, Damien Costa, Yonas Yimam Ayene, Ven-ceslas Villier, Abdelmounaim Mouhajir, Loic Favennec, Romy Razakandrainibe, Haileeysus Adamu.

**Project administration:** Ambachew W. Hailu, Abraham Degarege, Beyene Petros, Damien Costa, Loic Favennec, Romy Razakandrainibe, Haileeysus Adamu.

**Resources:** Ambachew W. Hailu, Abraham Degarege, Damien Costa, Loic Favennec.

**Software:** Ambachew W. Hailu, Abraham Degarege, Damien Costa, Yonas Yimam Ayene, Ven-ceslas Villier, Abdelmounaim Mouhajir, Romy Razakandrainibe.

**Supervision:** Ambachew W. Hailu, Abraham Degarege, Beyene Petros, Damien Costa, Ven-ceslas Villier, Loic Favennec, Romy Razakandrainibe, Haileeysus Adamu.

**Validation:** Ambachew W. Hailu, Abraham Degarege, Beyene Petros, Damien Costa, Ven-ceslas Villier, Loic Favennec, Romy Razakandrainibe, Haileeysus Adamu.

**Visualization:** Ambachew W. Hailu, Abraham Degarege, Beyene Petros, Damien Costa, Yonas Yimam Ayene, Ven-ceslas Villier, Abdelmounaim Mouhajir, Loic Favennec, Romy Razakandrainibe, Haileeysus Adamu.

**Writing – original draft:** Ambachew W. Hailu, Abraham Degarege, Yonas Yimam Ayene, Romy Razakandrainibe.

**Writing – review & editing:** Ambachew W. Hailu, Abraham Degarege, Yonas Yimam Ayene, Romy Razakandrainibe.

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
