## [Decision Letter · Decision Letter 0]

26 Jan 2022

PONE-D-21-37489Genetic diversity of Cryptosporidium spp. in non-human primates in rural and urban areas of EthiopiaPLOS ONE

Dear Dr. Hailu,

Thank you for submitting your manuscript to PLOS ONE. After careful consideration, we feel that it has merit but does not fully meet PLOS ONE’s publication criteria as it currently stands. Therefore, we invite you to submit a revised version of the manuscript that addresses the points raised during the review process.

We look forward to receiving your revised manuscript.

Kind regards,

Benjamin M. Rosenthal

Academic Editor

PLOS ONE

Journal Requirements:

3. Please include your tables as part of your main manuscript and remove the individual files. Please note that supplementary tables (should remain/ be uploaded) as separate "supporting information" files.

“NO”

6. We note that Figure 1 in your submission contain map images which may be copyrighted. All PLOS content is published under the Creative Commons Attribution License (CC BY 4.0), which means that the manuscript, images, and Supporting Information files will be freely available online, and any third party is permitted to access, download, copy, distribute, and use these materials in any way, even commercially, with proper attribution. For these reasons, we cannot publish previously copyrighted maps or satellite images created using proprietary data, such as Google software (Google Maps, Street View, and Earth). For more information, see our copyright guidelines: http://journals.plos.org/plosone/s/licenses-and-copyright.

7. Please upload a new copy of Figure 2 as the detail is not clear. Please follow the link for more information: https://blogs.plos.org/plos/2019/06/looking-good-tips-for-creating-your-plos-figures-graphics/

Reviewers' comments:

Reviewer's Responses to Questions

**Comments to the Author**

1. Is the manuscript technically sound, and do the data support the conclusions?

Reviewer #1: Partly

2. Has the statistical analysis been performed appropriately and rigorously? 

Reviewer #1: Yes

3. Have the authors made all data underlying the findings in their manuscript fully available?

Reviewer #1: Yes

4. Is the manuscript presented in an intelligible fashion and written in standard English?

Reviewer #1: No

5. Review Comments to the Author

Reviewer #1: The manuscript “Genetic diversity of Cryptosporidium spp. in non-human primates in rural and urban areas of Ethiopia” describes the results of a molecular survey of Cryptosporidium from two species of NHPs. The study is straightforward, methods seems appropriate, and the data from this study are of interest and value. However, the manuscript needs improvement in how the results have been presented and described before it is suitable for publication. My specific comments are as follows:

1. Data are from 2 species of NHP, Chlorocebus aethiops (vervet monkey) and Colobus guereza (mantled guereza). Methods seems to only describe collection of samples for vervet monkey. Also there are far more samples from vervet monkeys (177) than mantled guerezas (8). But description of results combines data from both species. Overall the manuscript would be improved by more clearly describing the data from each species included in the study.

2. Like for comment 1, descriptions of results and discussion of the data do not delineate between rural and urban populations in a clear manner. The manuscript would be improved by more clearly describing the data from the two sampling locations.

3. Results: when describing prevalence results, you describe a significant difference between males and females but don’t give the prevalence in each of these groups. Update text to include these values.

4. The tree does not seem to be necessary? It is not contributing to the results or discussion of the manuscript as data from the tree are not described or used to explain any aspects of this study.

5. What is the purpose of the last sentence of the results section? Those subtypes are described above. Why are they listed as additional here? I think this could be resolved by better describing data from each species and study site as suggested in comments 1 and 2.

6. In the discussion the language that parvum and hominis have broadened their hosts from humans to NHPs should be revised. As you point out the presence of these organisms in NHPs is not well surveyed at the molecular level, and you can not distinguish between shifts in host range versus observations in new hosts based on this study alone.

6. PLOS authors have the option to publish the peer review history of their article (what does this mean?). If published, this will include your full peer review and any attached files.

Reviewer #1: No

---

## [Author Response · Author response to Decision Letter 0]

10 Mar 2022

PONE-D-21-37489

Genetic diversity of Cryptosporidium spp. in non-human primates in rural and urban areas of Ethiopia

PLOS ONE

Dear Dr. Benjamin M. Rosenthal

Thank you for your comments and sending us the reviewer's comments. We thank also the reviewer for the constructive comments. We have revised the manuscript following the reviewer's suggestions. We described these changes in the below paragraphs. We hope that you will find our responses are acceptable, and we are looking forward to hearing your decision.

Editors 

Response: We have checked the formatting of the manuscript. The revised manuscript meets the Plos one’s style requirements. 

Response: We acknowledge the editor’s comment and included the text below in the methods section. 

“We performed non-invasive sampling [24] and collected only a fresh sample from the monkey's feces dripping off the floor. Verbal approval was obtained from the residents of Wurgissa and Hawassa, who live near the sampling site, and from local authorities (i.e., Hawassa city agriculture office and Wurgissa Kebele administration).” (See line 64-66 pages 5, 6)

3. Please include your tables as part of your main manuscript and remove the individual files. Please note that supplementary tables (should remain/ be uploaded) as separate "supporting information" files.

Response: We have included the table as part of the main manuscript in the revised submission.

“NO”

Response: This study didn’t receive funding support. So, we have included the text “The authors received no specific funding for this work” in the cover letter. 

You may need to state your data availability statement here and mention that you will specify this in the additional information section when you submit the revised manuscript.

Response: we have included data availability statement which reads ‘All relevant data are within the manuscript.” in the cover letter. We have also included the same statement as additional information during the online submission

6. We note that Figure 1 in your submission contain map images which may be copyrighted. All PLOS content is published under the Creative Commons Attribution License (CC BY 4.0), which means that the manuscript, images, and Supporting Information files will be freely available online, and any third party is permitted to access, download, copy, distribute, and use these materials in any way, even commercially, with proper attribution. For these reasons, we cannot publish previously copyrighted maps or satellite images created using proprietary data, such as Google software (Google Maps, Street View, and Earth). For more information, see our copyright guidelines: http://journals.plos.org/plosone/s/licenses-and-copyright.

Response: We acknowledge the editor’s suggestion and removed Fig 1 from the revised submission. 

7. Please upload a new copy of Figure 2 as the detail is not clear. Please follow the link for more information: https://blogs.plos.org/plos/2019/06/looking-good-tips-for-creating-your-plos-figures-graphics/. 

Response: We acknowledge the lack of clarity of the details in Fig 2, but we have removed the figure from the revised submission at the reviewer’s recommendation.

Response: We have checked the references to meet to Plos one’s style requirement. There are no retracted references but have included one new reference to the list (reference # 24).

Reviewers' comments: 

Comments to the Author

1. Is the manuscript technically sound, and do the data support the conclusions?

Reviewer #1: Partly

2. Has the statistical analysis been performed appropriately and rigorously?

Reviewer #1: Yes

3. Have the authors made all data underlying the findings in their manuscript fully available?

Reviewer #1: Yes

4. Is the manuscript presented in an intelligible fashion and written in standard English?

Reviewer #1: No

5. Review Comments to the Author

Reviewer #1

The manuscript “Genetic diversity of Cryptosporidium spp. in non-human primates in rural and urban areas of Ethiopia” describes the results of a molecular survey of Cryptosporidium from two species of NHPs. The study is straightforward, methods seems appropriate, and the data from this study are of interest and value. However, the manuscript needs improvement in how the results have been presented and described before it is suitable for publication. My specific comments are as follows:

1. Data are from 2 species of NHP, Chlorocebus aethiops (vervet monkey) and Colobus guereza (mantled guereza). Methods seems to only describe collection of samples for vervet monkey. Also there are far more samples from vervet monkeys (177) than mantled guerezas (8). But description of results combines data from both species. Overall, the manuscript would be improved by more clearly describing the data from each species included in the study.

Response: We have made clear that samples were collected from vervet monkey and mantled guereza in the method section and presented the data separately for the two species. We have copied below the added text in the methods and results section of the revised submission 

Method: 

“The study subjects were free-ranging wild vervet and colobus monkeys found in Hawassa town and the Vervet monkeys living in a rural village ‘Wurgissa’ when collecting the sample. Visitors, and people in the open recreation area of Hawassa town often feed the monkeys with leftover food. (see line 75-77; page 5 ) 

Results 

 “Out of 51 fecal samples tested positive for Cryptosporidium infection, 47 were obtained from Chlorocebus aethiops and 4 from Colobus guereza. ( see line 144-145 page 8) 

“Subtyping analysis of the C. parvum and C. hominis isolates from Chlorocebus aethiops identified five family subtypes for C. hominis {IaA20 (n=6), IaA26 (n=1); lbA10G2 (n=1), IdA21 (n=1) IeA11G3T3 (n=1)}, and six family subtypes for C. parvum {IIaA17G1R1 (n=6), IIaA19G2R1 (n=3), IIaA15G2R1(n=3) IIaA16G1R1 (n=1), IIaA17G2R1(n=1), IIaA20G1R1(n=1)}. Subtyping analysis of the four Cryptosporidium positive samples obtained from Colobus guereza revealed the subtype ‘IaA20’. ( see line 148-153 page 8 ) 

” 

2. Like for comment 1, descriptions of results and discussion of the data do not delineate between rural and urban populations in a clear manner. The manuscript would be improved by more clearly describing the data from the two sampling locations. 

Response: - We have presented and discussed the data stratifying it by the urban/rural regions where the sample was collected. “ 

Added text in the results and discussion 

Result

A significantly greater proportion of samples collected from monkey in urban (i.e. Hawassa, 42.5% ) area was positive for Cryptosporidium compared to samples obtained from monkey in the rural area (i.e., Wurgissa , 23.44%) (P= 0.017) (see page 8-9 line 127-130) 

A total of 12 subtypes { IIaA17G1R1(6), IIaA19G2R1(3), IIaA15G2R1(2), IIaA16G2R1(1) IIaA16G1R1(1) IIaA17G2R1(1), IIaA20G1R1(1), IaA20 (5), IdA21(1) lbA10G2(1), IeA11G3T3(1), IaA26(1) )} were seen from NHPs located in Rural Wurgissa but only two subtypes { IaA20(2), IIaA15G2R1 (1)}were identified from monkeys in the Hawassa town ( Table 2 ). (” See page 9 line 145 -149)

Discussion 

“A higher prevalence of Cryptosporidium infection was seen among NHPs living in the urban area compared to the NHPs living in the rural village. This difference might have resulted from ecologic factors (e.g., fecal contaminated water and close habitations), which may facilitate transmission of the parasite between NHPs and humans.” (See line 206-209, page 21).

3. Results: when describing prevalence results, you describe a significant difference between males and females but don’t give the prevalence in each of these groups. Update text to include these values. 

Response: We have provided prevalence data for males and females in the revised submission. It reads “The prevalence of Cryptosporidium infection was higher in female (35.4%) than male (14.70%) monkeys and the difference was statistically significant (p=0.002) (see line 124-126, page 11.”

4. The tree does not seem to be necessary? It is not contributing to the results or discussion of the manuscript as data from the tree are not described or used to explain any aspects of this study.

Response: - We acknowledge the reviewer’s comment and removed the phylogenetic tree from the revised manuscript. 

5. What is the purpose of the last sentence of the results section? Those subtypes are described above. Why are they listed as additional here? I think this could be resolved by better describing data from each species and study site as suggested in comments 1 and 2.

Response : - We have revised the text in the last paragraph of the result section. We have presented data stratifying by monkey spp. and region as described above.

The revised text in the last paragraph reads “ “ Subtyping analysis of the C. parvum and C. hominis isolates from Chlorocebus aethiops identified five family subtypes for C. hominis {IaA20 (n=6), IaA26 (n=1); lbA10G2 (n=1), IdA21 (n=1) IeA11G3T3 (n=1)}, and six family subtypes for C. parvum {IIaA17G1R1 (n=6), IIaA19G2R1 (n=3), IIaA15G2R1(n=3) IIaA16G1R1 (n=1), IIaA17G2R1(n=1), IIaA20G1R1(n=1)}. Only one family subtype (IaA20) was identified from Colobus guereza. A total of 12 subtypes {IIaA17G1R1(6), IIaA19G2R1(3), IIaA15G2R1(2), IIaA16G2R1(1) IIaA16G1R1(1) IIaA17G2R1(1), IIaA20G1R1(1), IaA20 (5), IdA21(1) lbA10G2(1), IeA11G3T3(1), IaA26(1) )} were seen from NHPs located in Rural Wurgissa but only two subtypes { IaA20(2), IIaA15G2R1 (1)}were identified from monkeys in the Hawassa town ( Table 2 ) (line 148-157, page8 ). 

6. In the discussion the language that parvum and hominis have broadened their hosts from humans to NHPs should be revised. As you point out the presence of these organisms in NHPs is not well surveyed at the molecular level, and you cannot distinguish between shifts in host range versus observations in new hosts based on this study alone.

Response: - We thank the reviewer for the suggestion. We have revised the text. It reads as “These reports suggest that C. hominis, which was initially described as infectious to humans, may potentially expand its hosts from humans to NHPs. (lines 181, 182 ; page 10)

6. PLOS authors have the option to publish the peer review history of their article (what does this mean?). If published, this will include your full peer review and any attached files.

Do you want your identity to be public for this peer review? For information about this choice, including consent withdrawal, please see our Privacy Policy.

Reviewer #1: No

---

## [Editor Report · Decision Letter 1]

4 Apr 2022

Genetic diversity of Cryptosporidium spp. in non-human primates in rural and urban areas of Ethiopia

PONE-D-21-37489R1

Dear Dr. Hailu,

We’re pleased to inform you that your manuscript has been judged scientifically suitable for publication and will be formally accepted for publication once it meets all outstanding technical requirements.

Kind regards,

Benjamin M. Rosenthal

Academic Editor

PLOS ONE
---

## [Editor Report · Acceptance letter]

7 Apr 2022

PONE-D-21-37489R1 

Genetic diversity of Cryptosporidium spp. in non-human primates in rural and urban areas of Ethiopia 

Dear Dr. Hailu:

I'm pleased to inform you that your manuscript has been deemed suitable for publication in PLOS ONE. Congratulations! Your manuscript is now with our production department. 

Kind regards, 

on behalf of

Dr. Benjamin M. Rosenthal 

Academic Editor

PLOS ONE